# On Improving the Training of Models for the Semantic Segmentation of Benthic Communities from Orthographic Imagery

**Gaia Pavoni [1],\***, **Massimiliano Corsini [1]**, **Marco Callieri [1]**, **Giuseppe Fiameni [2]**, **Clinton Edwards [3] and Paolo Cignoni [1]**

1 Visual Computing Lab (ISTI-CNR), 56124 Pisa, Italy; massimiliano.corsini@isti.cnr.it (M.C.); marco.callieri@isti.cnr.it (M.C.); paolo.cignoni@isti.cnr.it (P.C.)
2 NVIDIA AI Technology Centre (NVAITC), 40134 Bologna, Italy; gfiameni@nvidia.com
3 Scripps Institution of Oceanography, UC San Diego, La Jolla, CA 92037, USA; clint@ucsd.edu
* Correspondence: gaia.pavoni@isti.cnr.it

**Abstract:** The semantic segmentation of underwater imagery is an important step in the ecological analysis of coral habitats. To date, scientists produce fine-scale area annotations *manually*, an exceptionally time-consuming task that could be efficiently automatized by modern CNNs. This paper extends our previous work presented at the 3DUW'19 conference, outlining the workflow for the automated annotation of imagery from the first step of dataset preparation, to the last step of prediction reassembly. In particular, we propose an ecologically inspired strategy for an efficient dataset partition, an over-sampling methodology targeted on ortho-imagery, and a score fusion strategy. We also investigate the use of different loss functions in the optimization of a Deeplab V3+ model, to mitigate the class-imbalance problem and improve prediction accuracy on coral instance boundaries. The experimental results demonstrate the effectiveness of the ecologically inspired split in improving model performance, and quantify the advantages and limitations of the proposed over-sampling strategy. The extensive comparison of the loss functions gives numerous insights on the segmentation task; the Focal Tversky, typically used in the context of medical imaging (but not in remote sensing), results in the most convenient choice. By improving the accuracy of automated ortho image processing, the results presented here promise to meet the fundamental challenge of increasing the spatial and temporal scale of coral reef research, allowing researchers greater predictive ability to better manage coral reef resilience in the context of a changing environment.

**Keywords:** coral reef monitoring; orthomosaics; orthoprojections; semantic segmentation; deep learning

## 1. Introduction

*The Semantic Segmentation of Coral Reefs*

Large-area imaging is an increasingly popular solution for the study of subtidal environments at scales of tens to hundreds of meters. In particular, orthographic imagery, e.g., orthophotomosaics, orthoprojections (for brevity, **orthos**), is increasingly being used for spatio-temporal analysis of coral communities [1–5]. Orthos enable the fine-scale mapping and accurate measurements of coral colony size and position that allow researchers to extract the information to better understand the demographic patterns and the spatial dynamics of benthic communities. Previously, such information could only be obtained through laborious in situ methods, which necessarily limits the scale of the monitoring campaigns. Importantly, as large-area imaging allows researchers to conduct ecological data extraction

efforts digitally, researchers are now able to dramatically expand the spatial and temporal scales over which their work can be conducted. The challenge created, is thus in the efficient extraction of ecological information from large-area imagery.

To date, these fine-scale ecological maps are generated following a manual workflow, using GIS drawing tools or image painting software, such as Adobe Photoshop, to digitize individual coral colonies. In image analysis, the accurate identification of colonies boundaries and assignment of a taxonomic class, i.e., classification, is known as *semantic segmentation*. Unfortunately, manual segmentation of the coral population is an extremely time-consuming process, and since corals have a slow growth (linearly, less than a 1 cm/yr), the detection of change demands that segmentation be conducted with a high degree of accuracy. The manual workflow to extract information from large-area imagery has required up to an hour of human effort for each square meter of imagery, representing a severe bottleneck between the data collection and ecological analysis.

Driven by the trending research on Convolutional Neural Networks (CNN), deep learning approaches have become increasingly used for the automatic recognition of marine organisms. The seminal work on the automatic classification of benthic coral reef communities [6] led to the release of the Web-based platform for point annotations *CoralNet* [7]. However, point-based annotations of images are most valuable for determination of bulk metrics such as percent cover, and do not provide the data needed to quantify demographic change or to conduct spatial analyses: these investigations demand semantic segmentation. The optimization of a supervised model first requires the preparation of sizeable human-labeled datasets. In work by Alonso et al. 2017 [8], the authors propose a method to propagate annotations from point-based to pixel-wise, by manipulating the fluorescence channel. The resulting labeled masks are then used to fine-tune a SegNet [9] network. More recently, the same authors released the first extensive dataset of mask-labeled benthic images [10], obtained by the propagation of sparse annotation with a multi-level superpixel method. A custom annotation tool for the creation of segmented orthos of the seafloor, based on SLIC and graph-cut, is described in King [11]. The authors compared the performance of patch-based and semantic segmentation models, obtaining the best patch classification accuracy at 90.3% on ten coral taxa using a Resnet152 and the best pixel-wise accuracy of 67.70% with a Deeplab V2. In previous work [12], we fine-tuned a Deeplab V3+ using a large training dataset composed of ten orthos generated from Structure-from-Motion generated dense point clouds. The model outputs labeled maps with an accuracy comparable with human annotators for the binary segmentation task. Despite this progress, the automatic segmentation of sessile organisms in underwater environments still presents many challenges.

**Generalization.** There are over 800 known species of reef-building corals, *(Scleractinia)*, worldwide with many locations having hundreds of common species [13]. However, regional species pools vary widely [14], and corals are well known for their phenotypic plasticity and capacity for morphological adaptation to local conditions. The ideal classifier would classify every taxon present in a given ortho, independent of locality, conditions during image collection, or the equipment and software used in the construction of orthos. However, a CNN trained on one dataset composed of orthos from one location, with a given regionally adapted taxonomic assemblage (source domain), might learn domain features specific to that collection of species or region, and perform worse on new orthos with another collection of species (target domain).

Differences between domains might be due to the considerable morphological variability within and among marine species, the presence of survey/reconstruction-related features which affect ortho quality, as well as the variability of class abundance on different plots. Machine learning models usually expect a similar class frequency between source and target domain. However, the abundance of coral populations can vary widely from location to location, even within a single reef or island. Although it seems convenient to assume equally probable distributions during the CNN optimization, this might bias predictions on heavily unbalanced real-world scenarios.

**Class imbalance**. Surveyed areas often contain only a small number of individuals of the taxa under consideration, the *foreground* classes. The *background* class represents the remainder of the

benthos that is not actively targeted for monitoring, and includes everything from sand to less abundant coral taxa, and is often the predominant class in each input training tile. In our work to date, foreground class frequencies can vary from the 30 to 0.1% of pixels, resulting in mild to severe class imbalance. When a dataset contains many more samples of one class than the others, the classifier penalizes under-represented classes by focusing on the more numerous ones. Moreover, there is also considerable variation in colony size, both within and among taxa, with colony area ranging from a few squared centimeters to several squared meters. These sizes, in many cases, are also correlated with the specific class. A class containing only small coral instances, each being represented by a handful of pixels, will result in a class-imbalance scenario.

**Human errors.** Previous work [15] has shown that human recognition on photos reaches a variable accuracy on different benthic groups (hard coral, soft corals, algae): the average accuracy for hard corals is about $74 \pm 16\%$. In many cases, only in situ observations can disambiguate taxonomic classifications. Furthermore, humans do not recognize all coral classes equally well. To our experience, experts agree in identifying encrusting *Montipora* less than half of the time with respect to the other classes. This uncertainty in the annotations of training datasets in turn affects model reliability.

**Complex Borders.** The semantic segmentation of corals implies an accurate pixel-wise classification of the scene's objects. In CNN-based segmentation, networks typically generate output objects with smooth contours that do not reflect the jagged and sharp boundaries of coral frequently found in nature.The coarser resolution of predictions is partially due to the pyramidal structure of network architectures. Pooling layers combine high-level feature maps, downgrading the resolution and inducing a smoothing effect on outputs. As a result, most misclassification errors fall on instance boundaries. Further, the use of distribution-based loss functions, which do not take into account local properties, might aggravate this phenomenon. Finally, human annotation errors are also most likely to occur at colony boundaries, leading to inaccurate training datasets and inducing further ambiguity.

This paper is an extended version of our previous work [16], presented at the Underwater 3D Recording and Modeling conference. In that study, we investigated several strategies to improve the performance of CNN-based semantic segmentation of *a single coral class* on a *small*, problematic dataset from a single location, exploiting the properties of orthos. However, research and monitoring campaigns generally have multiple target classes and will generate orthos from several geographic locations of interest. Here, we extend the previous methodologies to a multi-class scenario.

The ecologically inspired splitting is a dataset preparation strategy dealing with the non-random coral spatial patterns in orthos [1,4]. As the spatial dispersion of coral populations is more descriptive than the simple class distribution, balancing according to this information improved the model accuracy both in training and testing.

Traditional class-imbalance learning techniques increase the importance of deriving information from under-represented classes to reduce learning bias towards the predominant classes. These methods are divided into two main categories, (i) those that act at the data level, essentially through under- or over-sampling, and (ii) those that act at the algorithm level, such as the adoption of dedicated cost functions. Interested readers may refer to Dong et al. 2018 [17] for an exhaustive overview of class-imbalance learning methodologies. Here, we tested different loss functions and then propose an over-sampling strategy based on Poisson-disk sampling. Some of the loss functions succeed in improving the accuracy on boundaries, while others better detect small coral instances. Finally, we re-propose a method to aggregate overlapping tile predictions using the prior information of the predicted coverage of the specimens on the surveyed area.

## 2. Materials and Methods

### 2.1. Materials

The dataset consists of human-labeled orthos from 10 plots collected by our team as part of the 100 Island Challenge project (http://100islandchallenge.org/) based at the Scripps Institution

of Oceanography, UC San Diego. The general approach used to create orthos has been presented elsewhere (refer to [5]) and will not be discussed in detail here. Briefly, at each island a series of 100 m$^2$ plots were imaged with Nikon d7000 cameras, capturing 2000–3000 highly overlapping images per plot, to create a single contiguous 3D model of each plot using the Structure-from-motion software Agisoft Metashape [18]. Point clouds were generated with "high" as the resolution parameter, and depth filtering was set to "mild". The dense cloud was then imported in the custom visualization platform *Viscore* [19] to create orthoprojections. Scale bars and ground control points were deployed in the field to provide scaling and orientation of the 3D model relative to the ocean surface, which is required for subsequent ortho-rectification.

In this study, we use orthos constructed from imagery collected in 2013 at Flint Island, Millennium Atoll, Vostok Island, Starbuck Island and Malden Island in the Southern Line Islands. At each island 1–3 100 m$^2$ permanent plots were established on the 10-m isobath on the leeward fore reef. Spanning over 500 miles from N-S, these geographically isolated islands offer an opportunity to refine algorithms in the context of local variation in species morphology, differences in size structure, and the overall and relative abundance of coral taxa.

The ten plots used in this work were manually annotated by experts using Adobe Photoshop CC, drawing per-pixel masks of six different classes, most including 2–3 morphologically and functionally similar species. The class *Pocillopora* includes several species from the genus *Pocillopora*, the class *Porites* (light-blue) includes three common massive species (*P. arnaudi*, *P. lobata* and *P. lutea*). The class plating *Montipora* (green) includes multiple plating species, and the class encrusting *Montipora* (olive) includes a suite of encrusting species, while the species *Montipora capitata* (light green) constitutes a separate class. The final class, *Background* (black), represents the remainder of the reef, including sand, algae, and other coral species. The average frequencies (e.g., percent cover of each class) of target classes in the whole dataset, over the ten orthos, are 55.72%, 8.1%, 17.0%, 11.1%, 4.26% and 3.81% for the *Background*, *Pocillopora*, plating *Montipora*, *Porites*, the encrusting *Montipora*, and *Montipora capitata* classes, respectively. There is a large variance between frequencies, while the background occupies about half of the seabed.

We measured the agreement between different annotators in detecting these target species using the Cohen's Kappa ($\kappa$). All classes were annotated with an excellent agreement ($\kappa > 0.8$) except for encrusting *Montipora* that has a poor/mediocre agreement ($0.4 < \kappa \leq 0.6$).

In Section 3.1, we substitute *Montipora capitata* with *Pocillopora grandis*, separating it from the rest of the Pocillopora genus, to test the effectiveness of the ecologically inspired dataset partition. For the same reason of experimenting with the over-sampling strategy, in Section 3.3 on the MIL-1 plot we also consider an additional class, *Corallimorph*.

## 2.2. An Ecologically Inspired Dataset Partition

In learning a specific task, the whole dataset should be a representative sample of the observation domain. In a classification setting is desirable that the training, validation, and test subsets contain a target class distribution similar to the entire dataset. Stratified sampling is a well-known technique in machine learning to avoid the creation of biased models.

Seafloor orthos are continuous spaces, where the marine organisms follow non-random patterns [1]. Figure 1 shows two labeled orthos: producing a manual partition into three representative sets of these plots is a non-trivial task.

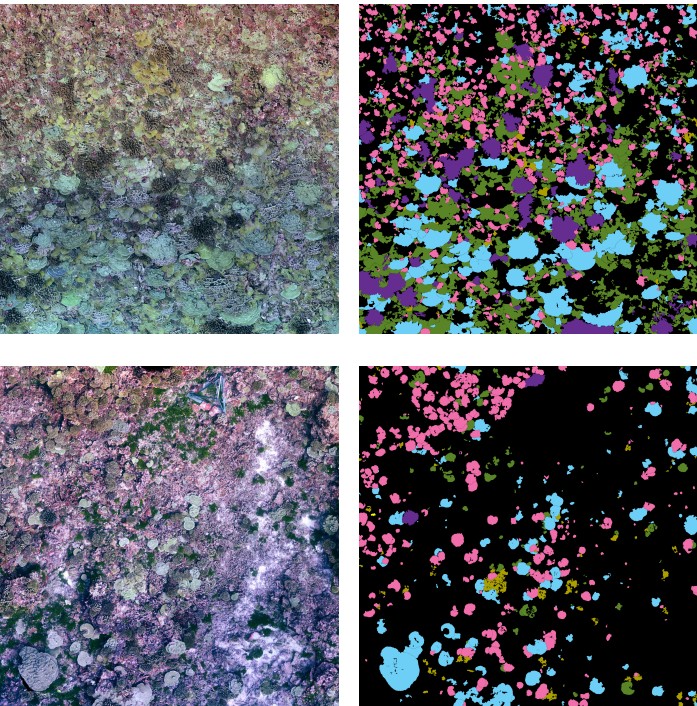

**Figure 1.** Orthoprojection and label map of the Fli-1 (**top row**) and Mil-4 (**bottom row**).

We propose to perform this partitioning *spatially*, by using demographic criteria. We measure the representativeness of the three sub-regions using some of the commonly used metrics of landscape ecology: *Patch Richness* of the corals, *Patch Coverage* of a given individual colony in the plot, and *Patch Size Coefficient of Variation (PSCV)*. The PSCV measures the standard deviation of the size of specimens as a percentage of the mean size all over the dataset:

$$PSCV = \frac{100 \cdot \text{std}(\text{Coral's areas on the Patch})}{\text{mean}(\text{Coral's areas on the Landscape})}. \tag{1}$$

The PSCV is commonly employed to describe the landscape area variability. The Patch Richness is the number of corals inside a patch, while the Patch Coverage is the density of the specimens. The density is calculated as:

$$Coverage(C, P) = \frac{P(C)}{\mathcal{A}_P} \tag{2}$$

where $P(C)$ is the number of pixels of class $C$ inside patch $P$, and $\mathcal{A}_P$ is the area of patch $P$ (in pixels). The selection algorithm generates multiple ($\sim$10,000) rectangular windows (patches) of a size approximately equal to the 15% of the ortho area and of different aspect ratios. Metrics are computed on each patch, and those with the best score of similarity with respect to the whole landscape are chosen as the validation and the test area, respectively. The training area is the remaining portion of the map. When two optimal windows cannot be found, they can be split into smaller sizes and then re-combined adequately to reach a more representative set.

The similarity score $S(C, P)$ for a patch $P$ and a class $C$, is:

$$S(C, P) = s_R(C, P) + s_C(C, P) + s_{PSCV}(C, P), \tag{3}$$

where:

$$
\begin{aligned}
s_R(C, P) &= ((Richness(C, P) - Richness(C, L))/Richness(C, L), \\
s_C(C, P) &= \|(Coverage(C, P) - Coverage(C, L))\|, \\
s_{PSCV}(C, P) &= \|PSCV(C, P) - PSVC(C, L)\|,
\end{aligned} \tag{4}
$$

and $L$ represents the Landscape. The three values $s_R$, $s_C$, and $s_{PSCV}$ are normalized using min-max normalization cause, usually, the values of PSCV are considerably higher than the others. In Section 3.1, we test the ecologically inspired splitting on plots with a non-random coral distribution.

## 2.3. Choosing a Loss Function

During network training, the minimization of the loss function progressively tunes the network weights, resulting in a fitting model. The loss function measures the discrepancies between predicted and actual values; how this difference is computed depends on the specific task. It is then possible, by changing how the loss function computes its value, to tackle different aspects and issues of the segmentation task. Loss functions in semantic segmentation belong to three main categories: *distribution-based*, *region-based*, and *boundary-based*.

Distribution-based losses measure the dissimilarity between probability distributions of random variables in the training set. The *Cross-Entropy* (CE) loss is the most widely used distribution-based loss; its popularity is due to the low complexity of the gradient calculation. The CE loss, by measuring distances between random variables, implicitly assumes comparable frequencies among the existing classes. This loss performs poorly in cases of class imbalance as it determines a decision boundaries bias in favor of the most represented class, leading to many false positives. The CE evaluates individual pixels, so highly represented classes contribute more to the loss computation with respect to the less represented ones.

The *cost-sensitive learning* is a technique employed in class-imbalance scenarios, and works by weighting the different class contributions in the loss function. When the positive predictions of the CE are weighted, it becomes the *Weighted Cross-Entropy* (WCE) [20,21]. The WCE gives more importance to under-represented classes; the inverse of class frequencies is a common choice of class weights. The model is encouraged not to miss classes with higher weights, decreasing the false-negative rate. Nevertheless, a rare class has few representative pixels, and the CE gradient calculated on a few pixels might be noisy. Weighting massively noisy values might cause instability during the training, so the weighted CE might fail to deal with severely unbalanced datasets.

The *Focal* loss (FL) [22] generalizes the CE for severe class-imbalance scenarios. By introducing a modulation factor, the FL assigns a lower importance to well-classified examples, forcing the model to focus on hard-negative examples (misclassified pixels). In the case of a severe class-imbalance (background pixels $\gg$ foreground pixels), misclassified pixels are mostly false negatives, so the use of FL aims at reducing them.

CE loss calculation outputs per-pixel values, there is no information related to the pixel position (i.e., whether it belongs to a border or an inner region). Therefore, they equally learn the classification of all pixels. However, without including contextual information, the boundaries of predicted segmentations appear uncertain. Region-based loss functions, measuring the mismatch between the predicted segmentation and the ground-truth labels, do not limit the calculation to single pixels but include the contextual information.

The standard region-based loss, the *Dice* loss (DL) [23] derives from the Dice coefficient, and it penalizes labels with a small overlap. The intersection between two masks can be expressed as an element-wise multiplication. Since they are ground-truth label binary masks, the multiplication eliminates those predicted pixels not belonging to the ground truth, cleaning up the boundaries. In the *Generalized Dice* loss (GDL) [24], the contribution of each class is corrected by weighting for the inverse of class-label frequencies.

The *Tversky* loss (TL) [25,26] balances the importance of false positives and false negatives, improving the trade-off between precision and recall through the introduction of two additional parameters. Finally, the *Focal Tversky* (FT) loss focuses on hard examples and performs better on highly imbalanced context.

Distributional and regional losses integrate predictions over the segmentation regions. In highly unbalanced scenarios, some target classes cover considerably smaller areas than other classes, affecting training stability. The Boundary loss [27], introduces a distance metric that sums contributions on shapes contours of target classes. Working on contours mitigates high class-imbalance scenarios and, at the same time, places more focus on the object profiles with respect to regional losses. Here, we propose to combine the Boundary loss with GDL to improve its convergence. The training of the network begins with the minimization of the mismatch between the predicted areas using the GDL and gradually passes to the minimization of the mismatch between the contours using the Boundary loss (GDL+Boundary Loss).

As mentioned, a particular challenge in the automatic segmentation of corals can result when the semantic classes are severely unbalanced, as is common, and the size of the objects varies widely, which can range from a handful, through to thousands of pixels in the orthos considered here. Furthermore, the shapes of the corals are complex, jagged shapes, and a large quantity of misclassified pixels falls on their outlines. Section 3.2 compares the performance of 4 different loss functions: the WCE, the GDL, the FT, and the GDL+Boundary, to understand which one best suits the coral semantic segmentation task.

## 2.4. Mitigating Imbalance with Poisson-Disk Based Over-Sampling

The absolute and relative abundance of a given coral genera can vary dramatically between geographical regions. Furthermore, while some coral species tend to be more "predominant" others more "rare", irrespective of location, these distributions may still be significantly different among regions, or even within a single plot. CNN models tend to reproduce the class probability distribution learned during training, constraining the creation of a semantic segmentation model applicable to each dataset. Traditional balancing strategies at a data level include the undersampling of the majority class or over-sampling of the minority classes. However, undersampling is often undesirable when working with complex image data. The *Background* class itself, which includes many different types of objects, from sand to non-target coral classes, can hardly be undersampled without losing information. In this work, we operate at data level, taking advantage of working with large images to avoid the simple repetition of the training data that might cause model's overfitting.

For a given ortho, we sample a set of CNN input tiles according to some reference frequencies. These frequencies, which should theoretically match the occurrence of the classes in the target domain, are not known *a priori*. In our case, we first calculate the frequencies on the entire dataset, composed of all ten maps, and then balance only the heavily unbalanced classes.

Our proposed sampling method is to crop out those input tiles with different local densities in such a way that the minority classes receive more samples. To do so, we choose the center of the tiles using the Poisson-disk sampling: a well-known sampling technique to generate random samples no closer than a specified minimum distance of $2r$, so a disk of radius $r$ centered on the sample never overlaps with the others (see Figure 2). One of the algorithms used to generate Poisson-disk samples is Dart Throwing. This algorithm generates samples checking if each new sample overlaps the disks centered on the existing ones. In the event that overlap occurs, the sample is discarded and a new one is generated.

Thus, we sample corals using the Dart Throwing algorithm with a class-dependent radius, to balance the number of pixels of the majority and minority classes. We calculate a per-class radius using a factor $K$, a multiplication factor required for each class to reach the number of pixels of the majority class. As the size of the input tiles is $513 \times 513$ pixels, the radius is approximately $256/\sqrt{K^\gamma}$ pixels. The $\gamma$ exponent is slightly greater than one ($\approx 1.3$) and it is used to decrease the radius for

rare coral classes (which typically, are also smaller in size). Despite this very rough approximation, according our experiments the method can improve balancing.

To guarantee that rare classes covering a limited number of pixels on the ortho receive the appropriate number of samples, we prioritize the sampling by starting from the least frequent class. In this way, we avoid covering a small coral belonging to a minority class with the disk centered on an adjacent coral. For the same reason, the *Background* class is sampled at the end. Following this procedure, oversampled dataset tiles are not simply duplicated; the local information (labeled pixels) does not change, but the per-tile contextual one does differ. Tests on this over-sampling technique are reported in Section 3.3.

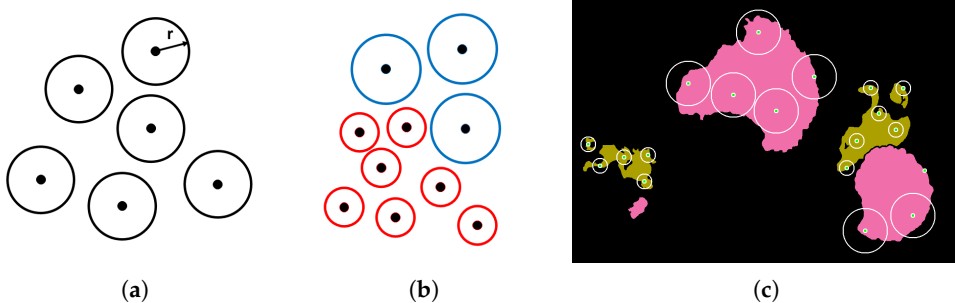

|     (a)     |     (b)     |     (c)     |

**Figure 2.** Ortho sampling strategy. (**a**) The Poisson-disk sampling generates random samples such that a disk of radius *r*, centered on each sample, never overlaps with other disks. (**b**) The radius changes such that a minority class (red) receives more samples than a majority one (blue). An example is shown in (**c**): the radius of the *Pocillopora* (pink) and encrusting *Montipora* (olive) classes are respectively 30 and 60 pixels.

### 2.5. Re-Assembling the Output of the CNN

In remote sensing applications, the size of a give ortho image file is usually too large to be processed entirely in a single pass due to GPU memory constraints. In these cases, the segmentation is applied to an overlapping sliding window to ensure the class consistency, in particular on a tile border. CNNs are translation-invariant; however, and a target object that is only partially visible on the tiles' extremes might be misclassified, introducing small but significant differences that require a fusion strategy when the tiles are re-combined in the final segmentation. A standard method to merge scores is to simply average the overlapping results [28,29]. Here, we propose to employ a method already used in multi-view stereo matching [30]. This method employs the Bayesian Fusion to aggregate the scores belonging to the same pixel.

Defining $S_n = \{s_1, s_2, \ldots, s_n\}$ as a set of classification scores for a given pixel, generated by the sliding window in different positions, according to the Bayes rule we can write:

$$p(y|S_n) = \frac{p(S_n|y, S_{n-1}) p(y|S_{n-1})}{p(S_n)}, \tag{5}$$

where *y* is the network output for a given pixel. By assuming that the scores are i.i.d:

$$p(y|S_n) = \mu p(y) \prod_{i=1}^{n} p(s_i|y), \tag{6}$$

where $\mu$ is a constant. Thus, the final Bayesian aggregation becomes:

$$p(y = c|S_n) \quad = \quad p(y = c) \prod_{i=1}^{n} p(s_i|y = c), \tag{7}$$

where $p(y = c|S_n)$ is the probability that the pixel belongs to the class $c$ given the scores $S_n$, $p(y = c)$ is the *a priori* probability of the class $c$, and $p(s_i|y = c)$ is the likelihood that a given class produces the score $s_i$. Please note that $p(y = c|S_n)$, to be valid probabilities, must be normalized to ensure that their sum across classes is 1.

The probability that a pixel of the target domain belongs to a certain class may be not known *a priori*. In this case, when the model performance is good enough, a reasonable estimation of the prior probabilities coincides with the predicted frequencies on the tiles before the fusion operation.

Section 3.4 shows the label map tiling artifacts and compares the Bayesian fusion with the standard average fusion.

## 2.6. Preprocessing and Training

Input orthos come from different reconstructions, each at slightly different scales. As the pixel size is a crucial information in the classification of corals, all orthos were first re-scaled at 1 px = 1.1 mm. Scaled plots are then sliced into large overlapping tiles of $1026 \times 1026$ pixels (scan order: left to right, top to bottom) sampling the tiles centers with a constant step of 513 pixels (regular sampling). The tiles for the tests without the ecologically inspired partition are split into training, validation and test sets considering the first 70% of consecutive tiles in the training set, the next 15% as the validation set, and the remaining 15% as the test set. We refer to this partition as *uniform partition*. The ten ortho dataset, with a uniform partition, includes 2430 tiles in the training dataset, 782 in the validation dataset, and 380 in the test dataset.

In Section 3.1, where we limit the study to 4 orthos with clustered coral classes, the regular sampling with the uniform partition generated an output dataset of 854 tiles for the training, 245 for the validation, and 245 for testing. The regular sampling on the ecologically inspired partition, instead, generates a dataset containing 750, 263, and 245 tiles in the three sets.

To improve domain adaptation, we employ some basic techniques such as colorimetric and geometric augmentation, weight decay and batch normalization. We perform geometric augmentation by replaced small random rotations and translations, but avoiding scale changes. Colorimetric augmentation simulates alterations related to the formation of underwater images (RGB color shift, light intensity variation, turbidity). After augmentation, tiles are center-cropped at a resolution of $513 \times 513$ pixels, which corresponds to the input image size of the CNN. The online preprocessing subtracts to each tile the dataset per-channel average value.

We choose the DeepLab V3+ [31] network, which uses separate atrous convolutional layers to ensure higher-resolution outputs (regarding the SegNet model proposed in [16]). The training follows a *transfer learning* approach, i.e., the network is initialized with pre-trained weights. Since corals have a very low visual similarity with images containing the every-day objects of PASCAL VOC 2012, all the parameters were let unfrozen. The fine-tuning learning rate was set lower than the one used during the training, to allow just small updates of weights and contrast the forgetting of high-level features. As optimizer, we use the Quasi-Hyperbolic Adam optimizer [32] with adaptive learning rate decay, an initial learning rate of $10^{-5}$, and a L2 penalty of $10^{-4}$. For each experiment, the network has been trained for 100–150 epochs (depending on whether the loss continued to decrease), on a NVIDIA GPU V100 with 32GB of memory, with a batch size of 64. Due to the large number of tiles (around 2400 in the training set), training time for one experiment was about 50 h on a single GPU. Considering that the loss function investigation needed several experiments (see Section 3.2), the set of experiments here reported required more than 1500 GPU hours.

## 3. Results

### 3.1. Ecologically Inspired Dataset Partition

To test the improvements brought by the ecologically inspired dataset partition, we selected four particularly challenging orthos, FLI-1, VOS-1, MIL-1, and MIL-4, displaying non-uniform coral class distributions. Next, we created two datasets, one using the uniform partition, the other using the ecologically inspired one. We then trained two models, and we compared the predicted mIoU on the test sets.

Figure 1 shows FLI-1 and MIL-1 original plots and labeled maps. In FLI-1, *Pocillopora* mostly occupies the upper part of the ortho, while *Porites* dominates the lower portion. In *MIL-4*, *Pocillopora* is concentrated on the top-left part of the ortho, while *Porites* is largely found on the bottom-left.

Figure 3 compares the label maps splitting according to different partitions. The first column shows the labeled map, the second column the uniform partition of the plot, and the third column the ecologically inspired partition into training (green), validation (blue) and test (red) subsets. Table 1 reports the landscape metrics (calculated on the entire plot) and the metrics calculated on validation areas of the four different plots: numbers are comparable. The similarity scores *S* of the validation area are 0.104, 0.139, 0.106, and 0.185, respectively, for FLI-1, MIL-1, MIL-4, and VOS-1 plots, indicating an ecologically similar distribution between the two sets.

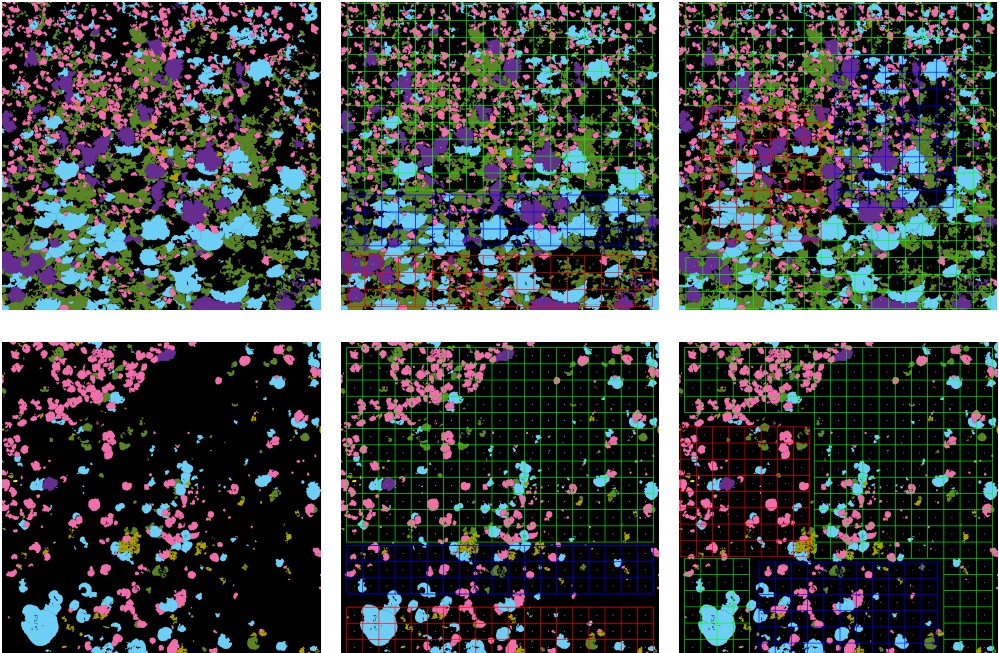

**Figure 3.** Results of the ecologically inspired partition of the orthos vs a uniform partition. The validation area is colored in blue, the test in red. The small rectangles represents the dataset clipped tiles. The first row is related to the labels of the FLI-1 plot, the second row to the MIL-4 plot.

We also compared results obtained by selecting the training/validation areas using the uniform or the ecologically inspired partition, keeping always as test area the red part at the bottom of the map, to favor the consistency of numerical results. Table 2 reports the accuracy and mIoU of the two models.

**Table 1.** Ecological metrics computed on the entire plot (Landscape) and the same metrics computed on the validation area found by our algorithm (Validation). Please note that the Richness of the validation region should be the 15% of the Landscape Richness because its area is the 15% of the area of the plot.

| Plot | Metric | Pocillopora | Pocillopora Grandis | Porites | Plating Montipora | Encrusting Montipora |
|------|--------|-------------|---------------------|---------|-------------------|----------------------|
| FLI-1 | Richness (Landscape) | 592 | 90 | 210 | 1017 | 63 |
| FLI-1 | Richness (Validation) | 101 | 11 | 26 | 155 | 8 |
| FLI-1 | Coverage (Landscape) | 10.597 | 7.653 | 14.712 | 20.007 | 0.665 |
| FLI-1 | Coverage (Validation) | 10.797 | 7.665 | 14.237 | 21.874 | 0.902 |
| FLI-1 | PSCV (Landscape) | 84.592 | 160.040 | 142.663 | 183.539 | 119.115 |
| FLI-1 | PSCV (Validation) | 84.486 | 142.159 | 119.417 | 158.965 | 112.291 |
| MIL-1 | Richness (Landscape) | 204 | 49 | 161 | 671 | 73 |
| MIL-1 | Richness (Validation) | 28 | 7 | 24 | 108 | 9 |
| MIL-1 | Coverage (Landscape) | 4.183 | 1.593 | 9.205 | 22.619 | 1.228 |
| MIL-1 | Coverage (Validation) | 4.831 | 2.165 | 3.894 | 22.959 | 0.677 |
| MIL-1 | PSCV (Landscape) | 108.486 | 190.7 | 213.948 | 242.031 | 227.171 |
| MIL-1 | PSCV (Validation) | 92.149 | 176.824 | 159.784 | 204.722 | 139.013 |
| MIL-4 | Richness (Landscape) | 299 | 2 | 122 | 89 | 81 |
| MIL-4 | Richness (Validation) | 38 | 0 | 19 | 13 | 11 |
| MIL-4 | Coverage (Landscape) | 8.531 | 0.309 | 6.411 | 1.544 | 1.077 |
| MIL-4 | Coverage (Validation) | 8.739 | 0.0 | 8.907 | 1.775 | 1.307 |
| MIL-4 | PSCV (Landscape) | 89.133 | 2.571 | 213.782 | 122.585 | 217.112 |
| MIL-4 | PSCV (Validation) | 86.209 | 0.0 | 177.363 | 105.917 | 83.992 |
| VOS-1 | Richness (Landscape) | 1002 | 3 | 199 | 202 | 147 |
| VOS-1 | Richness (Validation) | 154 | 1 | 28 | 32 | 22 |
| VOS-1 | Coverage (Landscape) | 14.535 | 0.107 | 19.722 | 9.041 | 3.219 |
| VOS-1 | Coverage (Validation) | 15.938 | 0.069 | 9.452 | 8.097 | 2.366 |
| VOS-1 | PSCV (Landscape) | 91.416 | 65.253 | 163.885 | 136.901 | 171.263 |
| VOS-1 | PSCV (Validation) | 92.968 | 0.0 | 167.097 | 127.981 | 117.155 |

**Table 2.** Model performance choosing the training/validation areas uniformly or according to the ecologically inspired partition. Test set is always the red area at the bottom of the map (see Figure 3).

| | Uniform Partition | Ecologically Inspired Partition |
|---|---|---|
| Training Accuracy | 0.920 | 0.942 |
| Validation Accuracy | 0.908 | 0.918 |
| Test Accuracy | 0.859 | 0.908 |
| Test mIoU | 0.762 | 0.836 |

*3.2. Comparison of Loss Functions*

Table 3 reports the performance of the Deeplab V3+ on the ten ortho dataset, minimizing different loss functions. The FT minimizing model achieves the best results, although all the models tested give acceptable performance. The low discrepancy between the accuracy of training and of validation does not reveal overfitting problems. Results with FT are slightly better, and this is supported by the qualitative analysis of the predicted labels (Figure 4).

**Table 3.** Model performance varying the loss function.

| | WCE | GDL | FT | GDL + Boundary |
|---|---|---|---|---|
| Training Accuracy | 0.930 | 0.939 | **0.951** | 0.945 |
| Validation Accuracy | 0.913 | 0.922 | **0.931** | 0.922 |
| Test Accuracy | 0.899 | 0.903 | **0.922** | 0.905 |
| Test mIoU | 0.821 | 0.825 | **0.85** | 0.830 |

Figure 4 qualitatively highlights the classifier's performance with respect to the choice of loss function, on a tile of the ortho FLI-1. Figure 5 compares the scores of the *Porites* and *Background* classes when the network minimizes the WCE and the FT. The region-based losses obtain prediction with less uncertainty and more contour-awareness. Predictions result in more confident label assignment both on contours and internal regions of each coral instance.

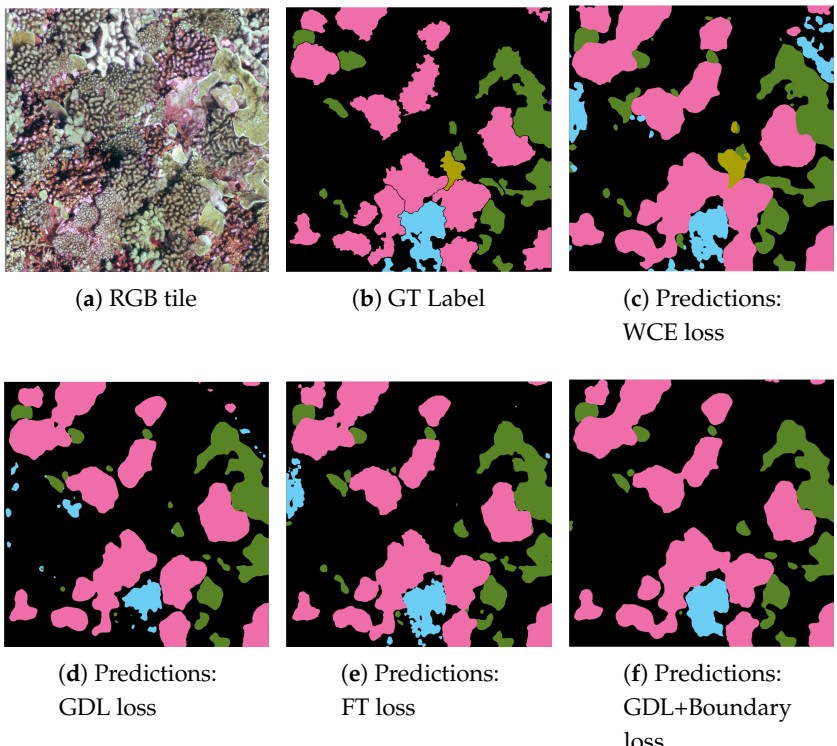

(**a**) RGB tile　　　　　(**b**) GT Label　　　　　(**c**) Predictions:
　　　　　　　　　　　　　　　　　　　　　　　　　　WCE loss

(**d**) Predictions:　　　　(**e**) Predictions:　　　　(**f**) Predictions:
GDL loss　　　　　　　　　FT loss　　　　　　　　　GDL+Boundary
　　　　　　　　　　　　　　　　　　　　　　　　　　loss

**Figure 4.** A dense coral coverage dataset tile (**a**) and the corresponding predicted labels (**b–f**). The Deeplab V3+ model has been fine-tuned using the same hyperparameters but minimizing different loss function. The model minimizing the WCE loss produces predictions with smoother contours and a high number of false positives. Nevertheless, the model that uses the WCE is the only one which correctly spots the colony of encrusting *Montipora*. The introduction of FT significantly improves the accuracy at instance boundaries. The use of GDL+Boundary loss penalizes the prediction of small colonies; some of the them are incorrectly classified as *Background*.

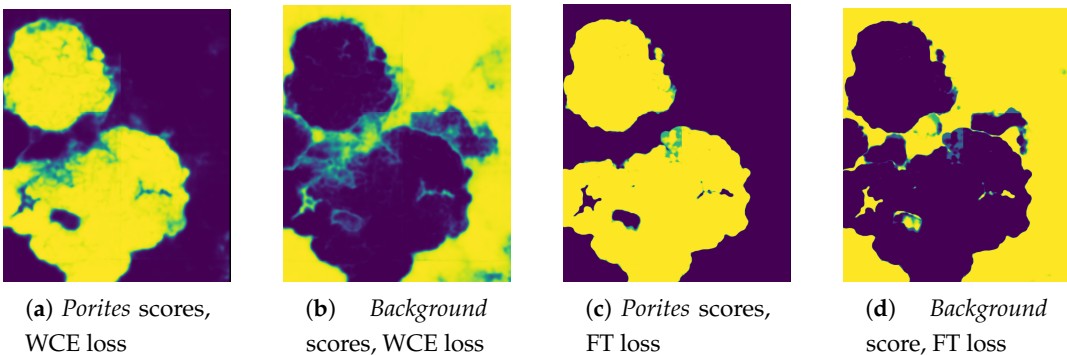

(**a**) *Porites* scores,　　(**b**)　*Background*　　(**c**) *Porites* scores,　　(**d**)　*Background*
WCE loss　　　　　　　scores, WCE loss　　　FT loss　　　　　　　score, FT loss

**Figure 5.** CNN scores for the *Porites* and *Background* classes: high probabilities are colored in yellow, low in purple, mid values in green-blue. As visible, the model optimized minimizing the FT exhibits less uncertain predictions.

Figure 6, compares the ground truth of the FLI-1 test area, with the labels predicted minimizing the WCE and with the FT. The accuracy and the mIoU values of the network using the two losses are respectively 0.899 and 0.819 for the WCE and 0.926 and 0.862 for the FT. Table 4 displays the mIoU per class. The pixel classification improves for each class except for encrusting *Montipora*, whose IoU is low when using WCE (0.165) and even lower when using FT (0.017).

**Table 4.** Annotations overlap (express as IoU) between GT and prediction on the FLI test area, Figure 6.

|  | *Pocillopora* | *Porites* | **Plating** *Montipora* | **Encrusting** *Montipora* | *Background* |
|---|---|---|---|---|---|
| WCE | 0.756 | 0.827 | 0.812 | 0.165 | 0.84 |
| FT | 0.814 | 0.879 | 0.838 | 0.017 | 0.885 |

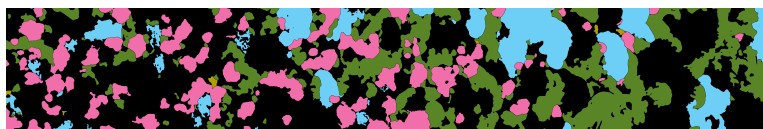

(**a**) Ground truth annotations.

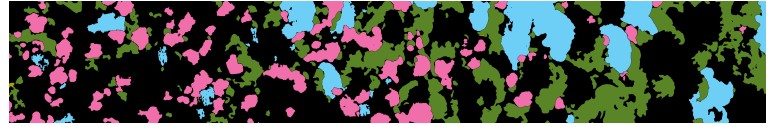

(**b**) Predicted annotations: CE loss.

(**c**) Predicted annotations: Focal Tvresky loss.

**Figure 6.** Automatic labeling generated from models minimizing different loss functions. Compared with GT (**a**), the model minimizing CE (**b**) outputs smother and larger regions. The use of FT (**c**) produces a definite improvement in the accuracy of the boundaries of all the detected classes; however, encrusting *Montipora*, a class with low frequency on the dataset, is almost entirely missed by FT.

In Figure 7, the use of WCE forces the network to predict positive pixels, increasing the number of false positives. False positives are concentrated around the edges of coral colonies due to the uncertain predictions on the boundaries of the model minimizing cross-entropy. The use of FT, as a region-based loss, creates labels with sharper contours that are more adherent to the coral shapes. However, it tends to increase the number of false negatives, decreasing the amount of predicted small areas belonging to minority classes.

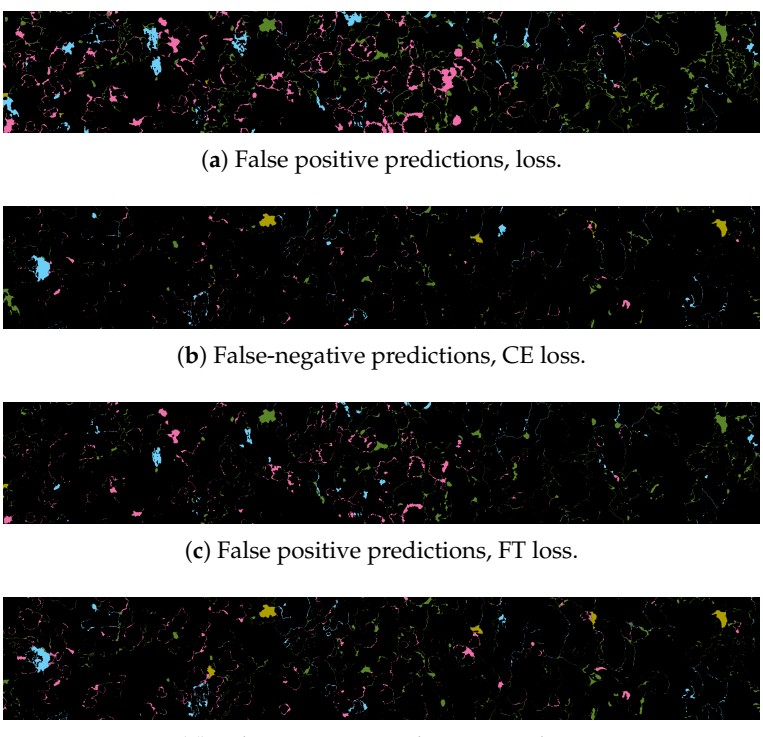

(**a**) False positive predictions, loss.

(**b**) False-negative predictions, CE loss.

(**c**) False positive predictions, FT loss.

(**d**) False-negative predictions, FT loss.

**Figure 7.** (**a**,**b**) The CE minimization outputs a model predicting a high number of false positive and labels with smoother shapes. (**c**,**d**) The FT loss model has a better balance between false positive and false negative, and produces shapes with sharper profiles. However, the encrusting *Montipora* colonies are almost entirely misclassified.

### 3.3. Poisson-Disk Over-Sampling

The massive use of over-sampling might bias predictions in the target domain, distorting the distributions of corals' populations. According to previous work [33], class rebalancing techniques improve the proportion of true positive predictions but negatively impact on model precision and accuracy. Hence, we decided to not alter too much the frequencies involved, keeping the percentage of the *Background* class high and adjusting only the severely imbalanced classes.

We started testing this sampling strategy by training a model for each of the ten orthos. We reported here two meaningful but opposite examples: MIL-1, where the balancing greatly improves the model performance, and MAI-1, where the balancing improvement is negligible (see Table 5).

**Table 5.** Class frequencies on training area before and after the application of our over-sampling strategy. The training area selection comes from the ecologically inspired partition.

| Plot | Background | Encrusting Montipora | Pocillopora Grandis | Porites | Plating Montipora | Pocillopora | Corallimorph |
|---|---|---|---|---|---|---|---|
| MIL-1 | 60.75% | 0.99% | 1.62% | 8.89% | 23.87% | 3.93% | - |
| MIL-1 (sampling) | 55.5% | 4.29% | 6.99% | 10.98% | 15.56% | 6.49% | - |
| Mai-1 | 83.61% | - | - | 12.32% | - | 2.95% | 1.12% |
| Mai-1 (sampling) | 80.7% | - | - | 11.93% | - | 5.46% | 5.46% |

The Deeplab V3+, trained on MIL-1 minimizing the WCE, reaches an accuracy of 0.799 and a mIoU of 0.714, while balancing the target classes, reaches an accuracy of 0.896 and a mIoU of 0.819. Figure 8 reports the training curves; Figure 9 shows the tiles produced by our sampling strategies.

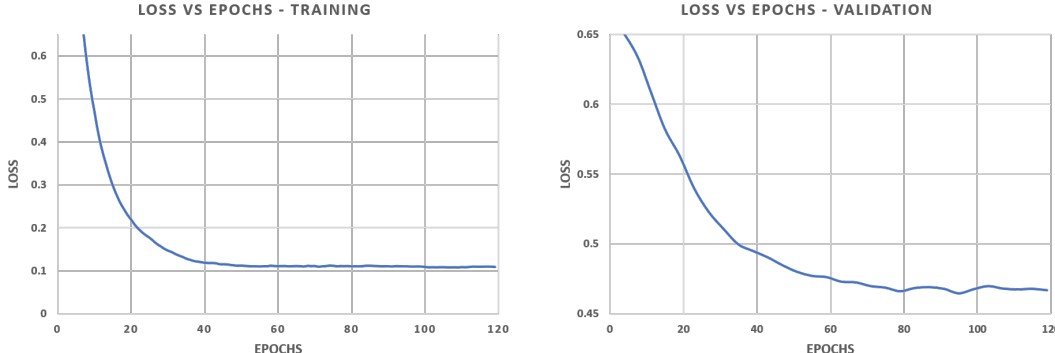

**Figure 8.** Loss plots for the oversampled (and ecologically partitioned) MIL-1. The curves trend does not highlight overfitting problems.

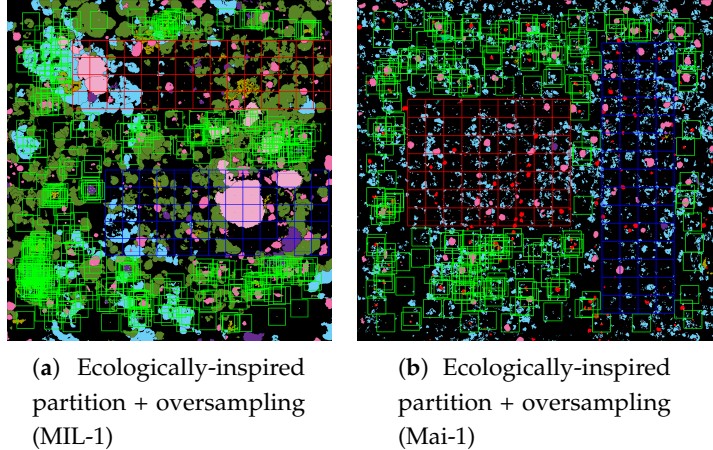

(**a**) Ecologically-inspired partition + oversampling (MIL-1)

(**b**) Ecologically-inspired partition + oversampling (Mai-1)

**Figure 9.** Oversampled tiles. (**a**) Ecologically inspired partition with oversampled training set on MIL-1. (**b**) Ecologically inspired partition with oversampled training set on MAI-1. Samples placed on the small-size corals incorporate a large amount of background pixels, preventing a good balancing of the classes.

We then applied the Poisson-Disk sampling to all ten orthos. The number of oversampled tiles depends on the corals' distribution; for this specific dataset, the number is only 14% more than the regular sampling. After calculating the overall frequencies of the 6 target classes, we sampled the ten orthos with the goal of balancing the most imbalanced classes, encrusting *Montipora* and *Montipora capitata*. With the Poisson-Disk sampling, the frequencies of *Background*, *Pocillopora*, plating *Montipora*, *Porites*, encrusting *Montipora*, and *Montipora capitata* change from 55.72%, 8.1%, 17.0%, 11.1%, 4.26%, and 3.81% to 50.39%, 8.58%, 11.95%, 10.91%, 9.15%, and 8.99%, respectively. Models trained on all 10 plots show no substantial improvement from the balance of encrusting *Montipora* and *Montipora capitata*. The model that optimizes the WCE earns a percentage point on the mIoU going from 0.821 to 0.830; on the other hand, the model that minimizes the FT loses it, going from 0.853 to 0.845. Table 6 compares the per-class IoU of the models trained with and without the over-sampling of encrusting *Montipora* and *Montipora capitata*.

**Table 6.** Annotation overlap (express as IoU) between GT and prediction on the FLI-1 test area, Figure 6. In this ortho, Poisson-disk sampling succeeds in improving the segmentation of all the classes.

|  | *Pocillopora* | *Porites* | *Plating Montipora* | *Encrusting Montipora* | *Background* |
|---|---|---|---|---|---|
| FT | 0.885 | 0.814 | 0.879 | 0.838 | 0.862 |
| FT/over-sampling | 0.893 | 0.818 | 0.901 | 0.845 | 0.872 |

*3.4. Bayesian Fusion for Tiles Re-Aggregation*

Table 7 reports the results of the tile aggregation strategies. Table 7 reports the results of the tile aggregation strategies. The model optimizing the FT loss infers predictions form HAW-1 tiles; tiles have a 50% overlap (see Section 2.6). Performances are numerically close; however, the tiling artifacts are corrected by the Average and the Bayesian fusion (Figure 10). Similar improvements are visible throughout all the plots.

**Table 7.** Classification results from aggregating scores of overlapping tiles (50% overlap in this case) using different methods.

| Method | Accuracy | mIoU |
|---|---|---|
| No fusion | 0.931 | 0.870 |
| Average | 0.931 | 0.872 |
| Bayesian fusion | 0.932 | 0.873 |

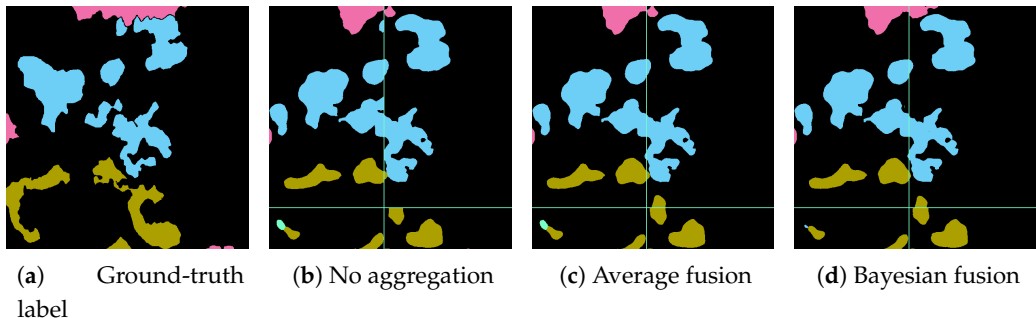

(**a**) Ground-truth label    (**b**) No aggregation    (**c**) Average fusion    (**d**) Bayesian fusion

**Figure 10.** Score aggregation across multiple overlapping tiles. (**a**) Ground-truth label. (**b**) No aggregation. (**c**) Average fusion. (**d**) Bayesian fusion. Without assembly, the classification results on tile borders (highlighted in light green) can be inconsistent, producing tiling artifacts (**b**). The Bayesian fusion (**d**), as well as the average aggregation (**c**), mitigate the problem making the final labeling closer to the ground truth (**a**). The prior knowledge of the predicted class frequencies does correct small classification errors. On the bottom-left of (**d**) there is a reduction of the light green class pixels.

## 4. Discussion

The ecologically inspired splitting is a simple and lightweight procedure that nevertheless may have a significant impact when applied on orthos with non-random coral patterns. As visible in Table 2, the ecologically inspired partition ensures higher model performance into both the training the test set. We emphasize that the improvement occurs even if the ecologically inspired partition contains 12% fewer of tiles for training. In small datasets characterized by imbalanced distributions, this technique reduces sampling bias on poorly represented classes, distributing the class features across the training, validation, and test sets.

We investigated the use of different loss functions in the model optimization. According to our results (and in agreement with the literature), the WCE predicts a higher number of positive cases, spotting more instances of rare classes. However, the model minimizing the WCE loss produces

predictions with smoother contours and an increased number of false positives. Nevertheless, the model that uses the WCE is the only one able to correctly spot the colonies of encrusting *Montipora*.

The FT minimization outputs a model producing a balanced number of a false positive and false-negative, labels with more accurate instance boundaries and a higher per-class IoU (Table 4). However, the FT model sometimes misses predicting some instances of rare classes (see Figure 7).

In the literature, the GDL+Boundary loss has been demonstrated to be useful, particularly in medical images, in predicting the segmentation of objects of similar sizes [27]. Conversely, in our tests, the GDL+Boundary loss achieved a lower mIoU w.r.t the FT, penalizing the prediction of small colonies, and displaying a smoother appearance of predicted masks. Our datasets contain corals with considerable variation in size, and the dependence of GDL+Boundary loss on the distance field for variably sized objects cause numerical instability during the model optimization.

Our datasets contain corals with considerable variation in size, and the dependence of GDL+Boundary loss on the distance field causes numerical instability during the model optimization for variably size objects.

The Poisson-Disk sampling methodology exploits the spatial continuity of orthos to balance class frequencies without causing overfitting, Figure 8. The modest results on MAI-1 highlight a limitation of this data balancing technique: orthos with a dense coverage of small corals (50–100 pixels) cannot be adequately balanced. The clipped tile centered on small instances includes an overabundance of non-target pixels, complicating the balancing. On the contrary, in MIL-1 the coral instances are more prominent and spaced out, and the balancing performs appropriately.

The overall performance of the oversampling on the dataset of 10 plots is almost unchanged regarding the uniform partition; new, correctly classified, pixels of encrusting *Montipora* and *Montipora capitata* represent only a small amount of the total. Additionally, encrusting *Montipora* and *Montipora capitata* are not present in the all maps. However, small regions undetected using uniform sampling are now correctly classified. In particular on FLI-1 (where *Montipora capitata* is missing) the results are visible in Table 6 and Figure 11. Figure 11 shows how this balancing recovers encrusting *Montipora* missing in Figure 4.

Bayesian fusion succeeds in outputting labeled maps without artifacts (Figure 10), correcting small clusters of misclassified pixels. Although the improvements are hardly appreciable from the numerical results (see Table 7), as this technique acts only on a small number of pixels on the total, re-assembled maps appear visually more coherent.

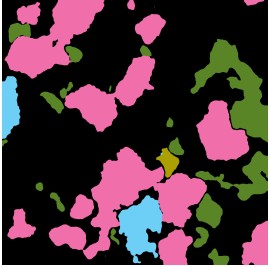

**Figure 11.** The model minimizing WCE does correctly classify encrusting *Montipora*. The same is true when minimizing FT, but only after a slight balancing of the rare classes using the Poisson-disk sampling. In this case, with respect to the model that is only minimizing WCE, the network is more accurate in predicting the contour pixels of colonies.

## 5. Conclusions and Future Work

This paper proposes several strategies to train a semantic segmentation CNN model on coral reef orthoprojections, from beginning (the dataset preparation) to end (the fusion of the model predictions). In Section 1, we highlighted some of the main challenges of the task: the generalization problem, the class-imbalance problem, and the low accuracy of automatic per-pixel predictions on coral

boundaries. We identified in the DeepLab V3+ the most suitable architecture to serve our purposes and Section 2.6, describes the training settings that we used to maximize the model generalization.

Randomly partitioning the input dataset into training, validation, and test sets without ensuring proper class representation might lead to unreliable results. Here, we proposed a sort of stratified learning approach by using an ecologically inspired dataset partition. According to presented results, this partition guarantees higher model performance both in the model training and in testing.

In the existing literature, it is suggested that the negative influence of class imbalance can be compensated for by working at the algorithm level or at the data level. We tried both by comparing the performance of different loss functions (cost-sensitive methods) and by proposing a data sampling strategy. We found that the model fine-tuned by minimizing the region-based Focal Tversky loss generates predictions with more accurate boundaries and a lower number of misclassified pixels, equally balanced in false positives and false negatives. Although the Focal Tversky loss demonstrates the best cost function to train coral segmentation models, it fails to deal with highly skewed datasets. We then proposed a data over-sampling method based on Poisson-Disk sampling that succeeded in improving predictions of rarest classes. However, class balancing is linked to the generalization problem, as frequencies in the target domain may vary significantly and equalizing them in the training dataset might worsen model performance.

Regarding the accuracy of coral colony boundaries, although the minimization of the FT loss demonstrated improvements to accuracy, the majority of the misclassified pixels still cluster on instance boundaries. We reserve to further investigate this point in future studies. In [21] the authors weigh contributions differently on coral boundaries using masks. Following a similar concept, in [34], the authors subdivide the background class into two separate sets with the instance boundaries proximity and weight them differently. Related work [35] proposes a custom combination of layer and loss to learn sharper and more adherent semantic boundaries. This flexible solution can be easily adapted to work with different semantic segmentation architecture. Recently, the GATED SCNN [36] demonstrated to output sharper predicted areas and achieves more robust performance on smaller objects.

The score aggregation eliminated the prediction artifacts over the border of input tiles. The Bayesian fusion, exploiting the predicted frequency information slightly improves the performance with respect to the traditional average fusion.

The performance of models in Section 3 is measured on the test areas of input orthos. However, the same models, when applied to totally unseen orthos from different geographical regions, and with different degrees of coral coverage, produce variable results. We performed several stress-test to evaluate the capabilities of the various models to generalize. On new orthos, the model's mIoU ranges from the 0.69 to the 0.97, depending on the species richness and abundance. In some target domains an entire coral genus can be misclassified if it happens to be visually similar to another which belongs to the background (and vice versa). We further reserve to explore other techniques for improving the model generalization in an unsupervised domain adaptations context. In the literature, there are several approaches to align frequencies between the source and target domains: [37] contains an exhaustive guide.

The use of a larger human-annotated dataset would contribute to improve the semantic segmentation model. Presently, there are several intelligent solutions to alleviate the human effort in manual labeling, facilitating the preparation of broader datasets. TagLab (https://github.com/cnr-isti-vclab/TagLab) is an AI-assisted annotation and analysis open-source software, specifically designed for the segmentation of benthic species. All the solutions presented in this paper have been implemented into TagLab.

**Author Contributions:** Conceptualization, G.P.; Data curation, C.E.: and G.P.; Methodology, G.P., M.C. (Massimiliano Corsini), M.C. (Marco Callieri); Formal analysis, G.P., M.C. (Massimiliano Corsini), M.C. (Marco Callieri), G.F.; Software, G.P., M.C. (Massimiliano Corsini), and G.F.; Validation, M.C. (Massimiliano Corsini) and G.F.; Investigation, G.P., M.C. (Massimiliano Corsini), G.F.; Resources, C.E.; Writing—original draft preparation, G.P., M.C. (Massimiliano Corsini), M.C. (Marco Callieri), G.F., and P.C.; Writing—review and editing, G.P., M.C. (Marco Callieri), M.C. (Massimiliano Corsini), G.F., C.E., and P.C.; Visualization, G.P., M.C. (Massimiliano Corsini), M.C. (Marco Callieri), and P.C.; Supervision, P.C. All authors have read and agreed to the published version of the manuscript.

**Funding:** This research received no external funding.

**Acknowledgments:** We gratefully acknowledge the NVIDIA AI Technology Center, EMEA, for its support and access to compute resources. Authors would like to thank the Sandin Lab (Scripps Institution of Oceanography, UCSD) for the collaboration and for kindly providing data.

**Conflicts of Interest:** The authors declare no conflict of interest.

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
