# Peer review of "On Improving the Training of Models for the Semantic Segmentation of Benthic Communities from Orthographic Imagery"

_remotesensing, doi:10.3390/rs12183106_

Round 1
Reviewer 1 Report
Review of "Semantic segmentation of benthic communities from ortho-mosaics"
General comments:
I believe this manuscript presents a very useful evaluation of modern techniques to improve the semantic segmentation of coral communities, which is field of study with much room for improvement and growth. These findings will be interesting and useful to readers attempting to use ML for this task (including current members of my lab!) or even similarly work such as tree crown species identification. While much of the methodology is scattered in multiple places and not clearly described in the Methods, I could gleam this information from the results section. From what I understand, the methods and results are straight-forward and properly interpreted. As such I have no real scientific criticism of the work done.
That said, the presentation of the work needs much improvement, falling a good distance from the suggested format from the Remote Sensing Author Instructions. This strongly affects the ability of readers to fully comprehend the study. I think significant refinement of the text is needed to get more directly to the intent of the work and the significance of the results.
Because the results may be interesting to readers, I recommend reconsideration of the manuscript after significant revision. The best thing to focus on in the revision is brevity. As part of this revision, a thorough edit to clean up the English is needed. There are a good number of mistakes (plural/singular, word choice, verb tense). They are generally small errors, but they do affect the understandbility of the text in several places.
Specific comments:
Title: This title gives no real indication of what is contained in the text. I suggest something more descriptive like "Overcoming challenges in semantic segmentation of benthic communities with modern machine learning innovations", or similar.
Abstract:
The abstract is pretty clear, but what were the significant results?
line 6: "outputs accurate area labeled maps" = output accurately- labelled maps?
Introduction:
The introduction is overly complex, and because of this it doesn't satisfactorily set the context for the methodology introduced later on. I suggest that the authors rewrite this section with a limited number of paragraphs that concisely introduce the need for classification of benthic images, describe the problem/challenges at hand (basically section 1.2, but condensed greatly), bring into light the more relevant work previously done in the field (including a brief version of section 2 and the earlier work by the same authors), then end with a paragraph describing specifically what will be addressed in the manuscript (similar to lines 83-86). This would prepare the reader to understand the methods being presented later on.
Related works:
I think some of this information should be incorporated into a Discussion that bring the related work into interpretation of the results. Some of this information can also be incorporated into a paragraph explaining the current state of benthic modelling. The rest of the information in this section is excessively detailed or unnecessary.
Methods:
I think the authors do a generally good job explaining the different metrics and algorithms and why they were selected here (although even here there is room for trimming extra information). What I don't see is enough detail about how the testing was done for each of the different data partitioning schemes, loss functions, the poisson disk vs random sampling approach, and the boundary segmentation methods. This is very much needed in this section.
Also, I cannot tell if the authors are claiming the poisson disk sampling approach as a novel tool of their own creation (line 289 makes is sound like this), but it has clearly been around a while. If the authors are suggesting their use of the sampling is novel, then this should be stated more clearly.
Results:
A lot of the methodology appears here in the first part of this section, but it is fairly general and not specific to any one part of the study. The full methodology for each part of the study needs to be defined in Methods. Also, the reader is often directed to look at a table without a sufficient of what results might be observed in the table.
Line 389: How were these validation areas selected?
Line 397-398: Can the authors describe what table 2 and figure 2 tell us here about the results? can repeat numbers in the tables and tell the reader how the methods differed with some detail.
Discussion:
Missing. Readers, especially those not deeply immersed in benthic stratification, would benefit from a good description of the results the authors found most noteworthy in context of other work and how the authors feel this work is significant to the community. I think some of the text from the Introduction, Related Works, and Conclusion could actually be pieced together to get a rough discussion to start with.
Conclusion:
This, too, needs to be much briefer, and authors should make sure that big results repeated here are also stated in the abstract.
Author Response
The authors sincerely thank the reviewer for the work and the suggestions on how to improve the manuscript.
REVIEWER 1
General comments:
I believe this manuscript presents a very useful evaluation of modern techniques to improve the semantic segmentation of coral communities,
which is field of study with much room for improvement and growth. These findings will be interesting and useful to readers attempting
to use ML for this task (including current members of my lab!) or even similarly work such as tree crown species identification.
While much of the methodology is scattered in multiple places and not clearly described in the Methods,
I could gleam this information from the results section. From what I understand, the methods and results are straight-forward and properly interpreted.
As such I have no real scientific criticism of the work done.
That said, the presentation of the work needs much improvement, falling a good distance from the suggested format from the Remote Sensing Author Instructions.
This strongly affects the ability of readers to fully comprehend the study. I think significant refinement of the text is needed to get more directly
to the intent of the work and the significance of the results.
Because the results may be interesting to readers, I recommend reconsideration of the manuscript after significant revision.
The best thing to focus on in the revision is brevity. As part of this revision, a thorough edit to clean up the English is needed.
There are a good number of mistakes (plural/singular, word choice, verb tense).
They are generally small errors, but they do affect the understandability of the text in several places.
ANSWER: We modify the manuscript to be compliant with the Remote Sensing format.
After the rewriting, the Methods contains the dataset and methods description, the Result section the experiments conducted and their numerical results, and the new section Discussion, the comments to the obtained results.
This should make the overall presentation more clear.
----------------------------------------------------------------------------------------------------
Specific comments:
Title: This title gives no real indication of what is contained in the text. I suggest something more descriptive like
"Overcoming challenges in semantic segmentation of benthic communities with modern machine learning innovations", or similar.
ANSWER: title has been changed to reflect the fact we are proposing new strategies to improve the training performance.
----------------------------------------------------------------------------------------------------
Abstract:
The abstract is pretty clear, but what were the significant results?
ANSWER: We have added the results obtained at the end of the abstract.
----------------------------------------------------------------------------------------------------
line 6: "outputs accurate area labeled maps" = output accurately- labelled maps?
ANSWER: rewritten
----------------------------------------------------------------------------------------------------
Introduction:
The introduction is overly complex, and because of this it doesn't satisfactorily set the context for the methodology introduced later on. I suggest that the authors rewrite this section with a limited number of paragraphs that concisely introduce the need for classification of benthic images, describe the problem/challenges at hand (basically section 1.2, but condensed greatly), bring into light the more relevant work previously done in the field (including a brief version of section 2 and the earlier work by the same authors), then end with a paragraph describing specifically what will be addressed in the manuscript (similar to lines 83-86). This would prepare the reader to understand the methods being presented later on.
ANSWER: We modify the Introduction improving the description of the context, something about the current state of the art, the open problems, and how the paper contributes to solving some of such problems.
----------------------------------------------------------------------------------------------------
Related works:
I think some of this information should be incorporated into a Discussion that bring the related work into interpretation of the results. Some of this information can also be incorporated into a paragraph explaining the current state of benthic modelling. The rest of the information in this section is excessively detailed or unnecessary.
ANSWER: We move parts of the Related Work in the Introduction to better assess the context and the motivation, and we remove the unnecessary parts.
----------------------------------------------------------------------------------------------------
Methods:
I think the authors do a generally good job explaining the different metrics and algorithms and why they were selected here (although even here there is room for trimming extra information). What I don't see is enough detail about how the testing was done for each of the different data partitioning schemes, loss functions, the poisson disk vs random sampling approach, and the boundary segmentation methods. This is very much needed in this section.
ANSWER: Thanks to the re-organization we have anticipated, in the Methods section, how the different methods are tested.
----------------------------------------------------------------------------------------------------
Also, I cannot tell if the authors are claiming the poisson disk sampling approach as a novel tool of their own creation (line 289 makes is sound like this), but it has clearly been around a while. If the authors are suggesting their use of the sampling is novel, then this should be stated more clearly.
ANSWER: The oversampling method based on cutting tiles according to a variable Poisson disk sampling distribution is an original contribution of us. The Poisson Disk sampling itself is a sampling technique. We have clarified this point in Section 2.4 ("The Poisson-disk sampling is a well-known sampling technique to generates..") and in the description of the contribution.
We also added some experiments in the Results section, and discussed them in the Discussion section, to better underline the advantages and the limits of the proposed oversampling strategy.
----------------------------------------------------------------------------------------------------
Results:
A lot of the methodology appears here in the first part of this section, but it is fairly general and not specific to any one part of the study. The full methodology for each part of the study needs to be defined in Methods. Also, the reader is often directed to look at a table without a sufficient of what results might be observed in the table.
ANSWER: It is true that some of the methodology presented is general, we have clarified some aspects and assess the advantages and disadvantages of the different techniques proposed in the Discussion. We have added detailed comments to the tables and the figures and improved the caption.
----------------------------------------------------------------------------------------------------
Line 389: How were these validation areas selected?
ANSWER: We have clarified in the text: The tiles for the tests without the ecologically-inspired partition are split into training, validation, and test sets by slicing the orthos from top to bottom: the first $70\%$ of consecutive tiles will be used as the training set, the next $15\%$ as the validation set, and the remaining $15\%$ as the test set. We will refer to this partition as the \emph{uniform partition}.
----------------------------------------------------------------------------------------------------
Line 397-398: Can the authors describe what table 2 and figure 2 tell us here about the results? can repeat numbers in the tables and tell the reader how the methods differed with some detail.
ANSWER: captions and their reference in the text have been expanded for most figures/tables
----------------------------------------------------------------------------------------------------
Discussion:
Missing. Readers, especially those not deeply immersed in benthic stratification, would benefit from a good description of the results the authors
found most noteworthy in the context of other work and how the authors feel this work is significant to the community.
I think some of the text from the Introduction, Related Works, and Conclusion could actually be pieced together to get a rough discussion to start with.
ANSWER: We have organized the paper as suggested.
----------------------------------------------------------------------------------------------------
Conclusion:
This, too, needs to be much briefer, and authors should make sure that big results repeated here are also stated in the abstract.
ANSWER: following the re-organization, moved most of the conclusion to the discussion. Added results to abstract
----------------------------------------------------------------------------------------------------
Reviewer 2 Report
This ms describes a method for automatic interpretation of reef coral assemblages and their cover by use of ortho-mosaics. The paper is well-written but the text could be shortened. The authors use data collected by others. There is no information on how the data was obtained (by drone?) and from where. It is also unclear how categories of corals were determined. Is this a subsample?
There are no acknowledgments and no statements on author’s contributions.
Abstract (lines 5, 9) should not contain citations because it also has a stand-alone function without reference list outside the published paper.
Line 4. Explain CNNs.
Lines 13-14. Keywords should no repeat words from the title. What is the proper spelling: “ortho-mosaics” in the title or “orthomosaics” as keyword? Orthoprojections -> Ortho-projections
Line 29. Suggested reference to explain neural networks: https://www.mdpi.com/1424-2818/12/1/29
Lines 31-32. Even the visual (in situ) recognition of sessile organisms in underwater environments can be a challenging task due to constraints in our taxonomic knowledge. So, what are minimum requirements for their automatic recognition?
Lines 60-61. Accuracy at which level (species, genera, categories of organisms)?
Lines 60-64. Explain whether you expect shortcomings caused by physical variables: depth (tidal fluctuations), refections caused by the moving seawater surface (waves, currents), suspended sediment (murkiness), angle of light penetration (position of the sun), slope of bottom. I cannot see how data were obtained
Line 66. A suitable source for an up-to-date number >800 of reef-bulding would bet his database: Hoeksema BW, Cairns S (2020). World List of Scleractinia. http://www.marinespecies.org/scleractinia/index.php
Line 87: A ecologically -> An ecologically
Line 133: machine learning methods -> machine-learning methods
Lines 133-134: underwater monitoring actions -> underwater-monitoring actions
Light 173: might varies -> might vary
Lines 173-175. You could support this information by references
Lines 338-344. This sounds like material and methods. I miss information about data collecting (how was the imagery produced? You also need to specify the areas from where the data was obtained. The species composition in the Atlantic differes from the Indo-West Pacific or East Pacific. So, it seems that your samples are from the Indo-Pacific.
Lines 341-344. Which are all the coral categories that were used? These seem to be some examples of many more categories. Or was only a selection of dominant species used?
Lines 343, 346, 365, 449, Fig 6, Fig 9, Table 5: Porite à Porites
Author Response
The authors sincerely thank the reviewer for the work and the suggestions on how to improve the manuscript.
REVIEWER 2
The paper is well-written but the text could be shortened.
ANSWER: We have shortened many parts, but we have also added new info requested by the reviewers
----------------------------------------------------------------------------------------------------
- The authors use data collected by others. There is no information on how the data was obtained (by drone?) and from where.
- Lines 338-344. This sounds like material and methods. I miss information about data collecting (how was the imagery produced? You also need to specify the areas where the data was obtained. The species composition in the Atlantic differs from the Indo-West Pacific or East Pacific. So, it seems that your samples are from the Indo-Pacific.
ANSWER: We have re-organized the paper to be compliant with the Remote Sensing format and to improve the quality of the overall presentation.
According to this re-organization, the section Methods contains, in the beginning, a description of the dataset and its acquisition (Section 2.1).
----------------------------------------------------------------------------------------------------
- It is also unclear how categories of corals were determined. Is this a subsample?
- Lines 341-344. Which are all the coral categories that were used? These seem to be some examples of many more categories. Or was only a selection of dominant species used?
ANSWER: Coral categories follow the World list of Scleractinia (see bibliography). The Target classes we used were those that had a coherent annotation across all the input maps and were the target of the monitoring for the project that collected these maps.
----------------------------------------------------------------------------------------------------
There are no acknowledgments and no statements on the author’s contributions.
ANSWER: We have added the Acknowledgement and the Author's contributions following the CREDIT system as requested by the Remote Sensing guidelines.
----------------------------------------------------------------------------------------------------
Abstract (lines 5, 9) should not contain citations because it also has a stand-alone function without a reference list outside the published paper.
ANSWER: We have removed all the citations from the abstract.
----------------------------------------------------------------------------------------------------
Line 4. Explain CNNs.
Line 29. Suggested reference to explain neural networks: https://www.mdpi.com/1424-2818/12/1/29
ANSWER: We have not included an explanation of the CNN because nowadays are becoming wide diffuse in the remote sensing field, but we have added the indicated reference in the Introduction.
----------------------------------------------------------------------------------------------------
Lines 13-14. Keywords should no-repeat words from the title. What is the proper spelling: “ortho-mosaics” in the title or “orthomosaics” as keyword? Orthoprojections -> Ortho-projections
ANSWER: We have uniformed the terms.
----------------------------------------------------------------------------------------------------
Lines 31-32. Even the visual (in situ) recognition of sessile organisms in underwater environments can be a challenging task due to constraints in our taxonomic knowledge. So, what are minimum requirements for their automatic recognition?
ANSWER: It is difficult to say, in general. The problem is not the CNN capabilities, but the quality of the input annotation. As human recognition is difficult, the labeling used in training will be incoherent preventing obtaining a classifier that has good recognition performance. The quality of the input annotations of the dataset used is reported in Section 2.1 (high agreement between the experts for all classes, save one).
----------------------------------------------------------------------------------------------------
Lines 60-61. Accuracy at which level (species, genera, categories of organisms)?
ANSWER: clarified in the text
----------------------------------------------------------------------------------------------------
Lines 60-64. Explain whether you expect shortcomings caused by physical variables: depth (tidal fluctuations), reflections caused by the moving seawater surface (waves, currents), suspended sediment (murkiness), angle of light penetration (position of the sun), slope of bottom. I cannot see how data were obtained
ANSWER: this falls somehow outside the scope of the work. Some of these artifacts are reduced by the process of creating the map (e.g. the process of creating the 3D point cloud does reduce lighting incoherence) what remains in the input maps should be addressed by the data augmentation step, described in Section 2.6, as we would like to train the model in recognizing objects independently by water conditions as the turbidity and the color shift.
----------------------------------------------------------------------------------------------------
Line 66. A suitable source for an up-to-date number >800 of reef-building would bet his database: Hoeksema BW, Cairns S (2020). World List of Scleractinia. http://www.marinespecies.org/scleractinia/index.php
ANSWER: Thank you, we have added it
----------------------------------------------------------------------------------------------------
Line 87: A ecologically -> An ecologically
Line 133: machine learning methods -> machine-learning methods
Lines 133-134: underwater monitoring actions -> underwater-monitoring actions
Light 173: might varies -> might vary
Lines 343, 346, 365, 449, Fig 6, Fig 9, Table 5: Porite à Porites
ANSWER: errors have been corrected, language revised, many parts have been rewritten
----------------------------------------------------------------------------------------------------
Lines 173-175. You could support this information by references
ANSWER: We deleted this sentence.
----------------------------------------------------------------------------------------------------
Round 2
Reviewer 1 Report
The authors did a good job cleaning up the presentation of the manuscript from the previous version. I feel the introduction puts the study in context very well, and a reader could now easily search for a particular methodology and quickly look up the results they are most interested in.
One thing that is missing is an explicit description of how the number of samples for training, validation, and test sets is determined. The authors explain how the patches that go into each set are selected, but I cannot find an account of these numbers anywhere except in a vague statement that there are about 2400 in the training set.
There are a few remaining grammatical errors, but these were reduced from the last version as well, not interfering with a reader's understanding. Some specific edits I had time to record:
Introduction -
Line 25-26: "... limited the scale over." is unclear.
Line 58. No need for a linebreak here.
Line 63: Change commons to common.
Line 95-96: "resulted in the 74+/ 16%" is missing some text.
Line 113: "Although this effort successful" add a verb.
Line 118: "demonstrates to improve" - rephrase
Line 120: Change increases to increase
Methods -
Line 138: Remove "are"
Line 163: Change well-know to well-known
Author Response
Dear Reviewer,
We want to thank you for your work. We corrected the language mistakes you reported, and we improved the lexicon throughout the entire paper.
We added the paragraphs highlighted in blue, where we have answered your question about the dataset size and creation. We believe that with this correction, the methodology will be more clear to readers.
Best regards,
The Authors.